# Make Your Decision Convincing! A Unified Two-Stage Framework: Self-Attribution and Decision-Making

**Yanrui Du, Sendong Zhao,**[*] **Haochun Wang, Yuhan Chen, Rui Bai**
**Zewen Qiang, Muzhen Cai, Bing Qin**
Harbin Institute of Technology, Harbin, China
{ yrdu, sdzhao, hcwang, yhchen, rbai, zwqiang, mzcai,qinb}@ir.hit.edu.cn

## Abstract

Explaining black-box model behavior with natural language has achieved impressive results in various NLP tasks. Recent research has explored the utilization of subsequences from the input text as a rationale, providing users with evidence to support the model decision. Although existing frameworks excel in generating high-quality rationales while achieving high task performance, they neglect to account for the unreliable link between the generated rationale and model decision. In simpler terms, a model may make correct decisions while attributing wrong rationales, or make poor decisions while attributing correct rationales. To mitigate this issue, we propose a unified two-stage framework known as Self-Attribution and Decision-Making (**SADM**). Through extensive experiments on five reasoning datasets from the ERASER benchmark, we demonstrate that our framework not only establishes a more reliable link between the generated rationale and model decision but also achieves competitive results in task performance and the quality of rationale. Furthermore, we explore the potential of our framework in semi-supervised scenarios.

## 1 Introduction

Large-scale pre-trained models (Lewis et al., 2019; Touvron et al., 2023) have achieved state-of-the-art results on various tasks (Wang et al., 2023; Du et al., 2023a; Chen et al., 2023), but their decision-making process is opaque. Recent work (Geirhos et al., 2020; Lai et al., 2021; Wang et al., 2021; Du et al., 2022; Liu et al., 2023; Du et al., 2023b) have revealed that models often rely on superficial clues for predictions, which can make their decisions unconvincing. Therefore, it is valuable to motivate models to provide trustworthy rationales to back up their decisions, which facilitates their implementation in real-world applications.

---

[*] Corresponding author

*Claim*: **The Nice Guys was directed by Stephen Spilberg.**

*Passage*: *The Nice Guys is a 2016 American neo-noir action comedy film directed by Shane Black and written by Black and Anthony Bagarozzi .* **... The Nice Guys premiered on May 15 , 2016 , at the 2016 Cannes Film Festival and was released by Warner Bros. …**

*Gold Answer*: **REFUTE.**

*Model Output*: **Answer: SUPPORT. Rationale: The Nice Guys is a 2016 American neo-noir action comedy film directed by Shane Black and written by Black and Anthony Bagarozzi .**

---

*Claim*: **The Others (2001 film) won awards.**

*Passage*: **The Others -LRB- Los Otros -RRB- is a 2001 Spanish-American supernatural gothic horror film with elements of psychological horror. …** *The Others was nominated for Saturn Awards including Best Director and Best Writing for Amenábar and Best Performance by a Younger Actor for Alakina Mann .* **…**

*Gold Answer*: **SUPPORT.**

*Model Output*: **Answer: SUPPORT. Rationale: The Others -LRB- Los Otros -RRB- is a 2001 Spanish-American supernatural gothic horror film with elements of psychological horror .**

Figure 1: With the claim and passage as input, the FID-Ex framework generates the model decision, followed by the rationale as output. It is observed that the generated rationale does not convincingly justify the model decision. The underlined part of the passage represents the manually annotated rationale.

Recent studies (Ismail et al., 2021; Shen et al., 2022) have concentrated on the ERASER (DeYoung et al., 2019) benchmark, which encourages models to obtain the extracted subsequences from the input text as a rationale to support their decision. The WT5 (Narang et al., 2020) framework and its variant FID-Ex (Lakhotia et al., 2020) have shown superiority on this benchmark, which generates the rationale and classification decision in a parallel way. However, such parallel frameworks raise a serious problem: is the link between the rationale and classification decision generated by models reliable? In the upper example of Fig. 1, we observe that the model attributes the correct rationale, that

"The Nice Guys is a 2016 American neo-noir action comedy film directed by Shane Black", but still mistakenly supports the claim that "The Nice Guys was directed by Stephen Spielberg". In the lower example of Fig. 1, it is evident that the model attributes the rationale that "The Others (Los Otros) is a 2001 Spanish-American supernatural gothic horror film with elements of psychological horror", which is entirely unrelated to the claim. However, despite the wrong rationale, the model still correctly supports the claim that "The Others (2001 film) won awards". These instances highlight a significant challenge in developing a model with explanations in natural language form, that the generated rationale does not genuinely support and convincingly justify the model decision.

To mitigate the above issue, we introduce a unified two-stage framework called Self-Attribution and Decision-Making (**SADM**). Our SADM framework adopts distinct architectures for training and inference processes. For the training process, we train the model by jointly optimizing both the self-attribution and decision-making objectives. For the inference process, we adopt a two-stage format, which is inspired by the two-stage inference theory of human (Evans, 1984). The model is first prompted to extract the rationale from the given input (known as the self-attribution stage), and then, the model is prompted to utilize the extracted rationale to make informed decisions (known as the decision-making stage). Moreover, our SADM framework incorporates the Fusion-In-Decoder (FID) (Izacard and Grave, 2020) architecture to address the challenges posed by lengthy texts, and the Sentence Mark (SM) (Lakhotia et al., 2020) strategy to mitigate the issue of random and irrelevant rationale generation during the self-attribution stage. To further enhance the model's comprehension at the decision-making stage, we also introduce a Reasoning Augment Learning (RAL) strategy.

In our experiments, we introduce the RSQ metric to quantitatively assess the reliable link between generated rationales and model decisions. Experimental results consistently show significant improvements in the RSQ metric for our SADM framework. For task performance and the quality of rationale, our SADM framework also outperforms strong baselines overall. Moreover, we conduct ablation experiments to analyze the contribution of each component within our framework.

## 2 Background

### 2.1 Task Form

We work with supervised data containing quadruples $(q, p, r, y)$. Here, $q$ represents a question or claim, $p$ represents a passage that can answer or judge $q$, $r$ corresponds to the rationale, typically a subsequence extracted from the passage $p$, and $y$ represents the classification target. Our objective is to train a model $f$ where the input is denoted as $x = (q, p)$. The desired outcome from the model is twofold: a classification result and a rationale to explain its decision-making behavior, that is, $(r, y) = f(q, p)$.

### 2.2 Related Work

**Rationale.** The rationale is defined as the condensed and logically coherent subsequences from the input text, yet still adequate for the model to make correct decisions (Lei et al., 2016; Linardatos et al., 2020; Burkart and Huber, 2021). Previous works (McDonnell et al., 2016, 2017; Arous et al., 2021) have shown that rationales bring many benefits, including more reliable judgments, greater transparency for evaluating human raters, and added value from the rationales themselves.

**Methods.** Existing methods are mainly divided into two categories: pipeline-based frameworks and parallel frameworks.

For pipeline-based frameworks, one way is a post-hoc explanation. After models make decisions, humans attempt to analyze why models give a specific decision. Common methods include attention mechanism (Tenney et al., 2019) (assign soft weights to tokens by self-attention matrix), LIME (Ribeiro et al., 2016) (approximate model behavior locally by repeatedly perturbing inputs), and gradient (Sundararajan et al., 2017; Smilkov et al., 2017) (gradient of each token vector to represent their attribution), etc. However, recent work (Feng et al., 2018; Serrano and Smith, 2019; Jain and Wallace, 2019; Pruthi et al., 2019; Brunner et al., 2019; Zhou et al., 2022) have pointed out that the above methods often exhibit counterintuitive behaviors and lack credibility. The other way is a pre-hoc explanation. Models are encouraged to first generate the rationale and then make decisions based on it. The BERT2BERT framework utilizes two independent models to extract rationales and make decisions, with the reparameterization method to jointly optimize two models during training. Based on the

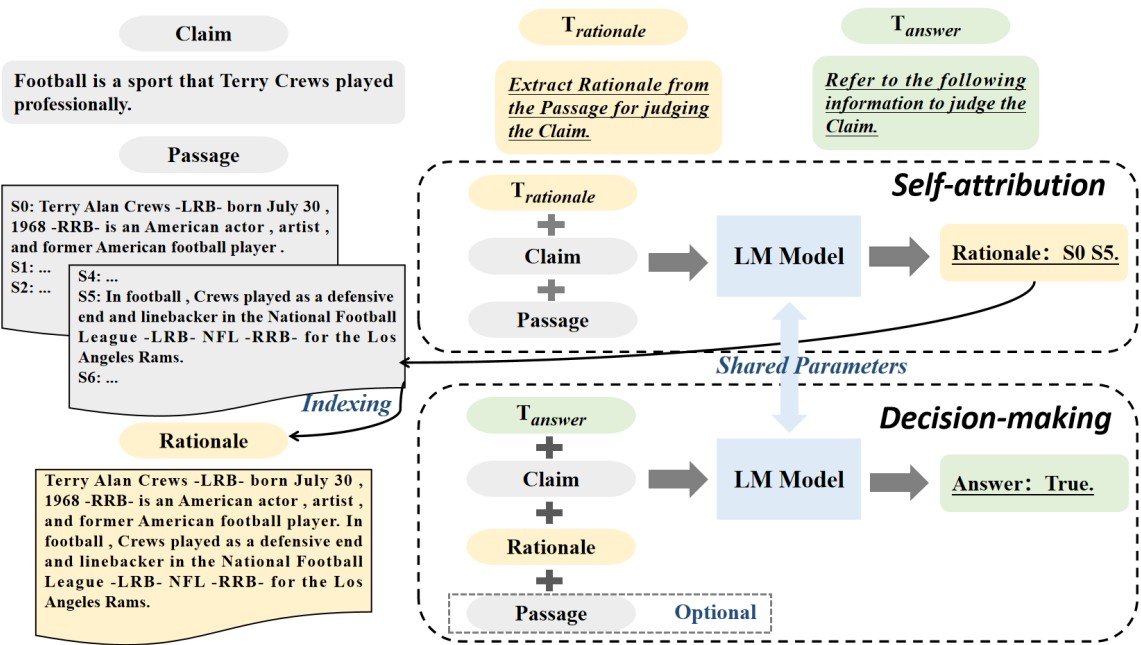

Figure 2: Illustration of our SADM framework: During the inference process, a two-stage format is adopted. Firstly, the model is prompted to generate the rationale, and subsequently, the model is prompted to make a decision based on the generated rationale.

BERT2BERT framework, the IB framework proposes an information bottleneck method instead of the reparameterization method to jointly optimize the models. Moreover, the QUASER framework incorporates a sentence selector and a multi-task training objective to improve the model's task performance, which aims at the sequence-to-sequence (seq2seq) models.

For parallel frameworks, WT5 and FID-Ex are representative, where the latter is a variant of the former. For the WT5 framework, task-specific phrases such as "explain fact verification task" are prepended to the input text. The model then generates a classification decision, followed by the extracted rationale. Notably, the position of the classification decision and extracted rationale can be interchanged, referred to as WT5-INVERSE (WT5-INV). Empirical evidence suggests that WT5 generally outperforms WT5-INV in terms of performance. Furthermore, FID-Ex retains the same mode as WT5 but introduces the fusion-in-decoder architecture to address input length limitations. Additionally, it employs a sentence mark strategy to prevent the generation of random rationales.

Furthermore, prior research (Wiegreffe et al., 2020) has measured the association between free-text rationales and model decisions. Their findings indicate that the parallel frameworks offer substantial advantages over the pipeline framework. Differently, our study concentrates on extracted rationale scenarios. In comparison to parallel frameworks, our proposed pipeline SADM framework demonstrates more pronounced advantages.

## 3 SADM

In this section, we first introduce our overall SADM framework (Sec. 3.1), and then we describe how our SADM framework works in detail from two aspects: Training Process (Sec. 3.2) and Inference Process (Sec. 3.3).

### 3.1 Overall Framework

The theory of human inference (Evans, 1984) suggests that there are two processes involved in human inference: the heuristic process and the analytic process. During the heuristic process, individuals gather task-relevant information, while the analytic process involves manipulating and processing the gathered information to make judgments. Drawing inspiration from this cognitive thinking, we propose our SADM framework, as depicted in Fig. 2. Firstly, we employ the $T_{rationale}$ template to prompt the model to generate a task-related rationale, the process called **self-attribution**. Subsequently, we employ the other $T_{answer}$ template to prompt the model to make a decision based on the generated rationale, the process called **decision-making**. As for the choice of prompt templates,

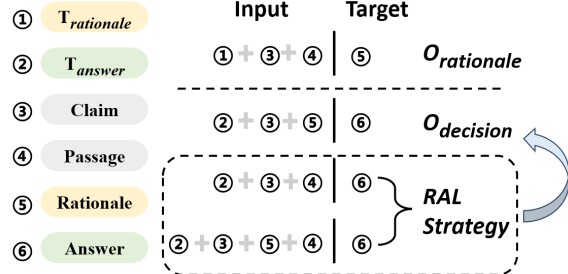

Figure 3: The format of training samples with and without the RAL strategy.

we utilize natural language oriented toward human understanding. Consider the FEVER dataset as an example. For the $T_{rationale}$ template, we design it as follows: "Extract the rationale from the passage to assess the claim". Similarly, the $T_{answer}$ template is formulated as: "Refer to the following information to judge the claim". Notably, we use discrete prompt templates, which do not introduce additional parameters or training costs.

Moreover, similar to FID-Ex and QUASER frameworks, we implement the FID architecture based on the seq2seq models to effectively handle lengthy text inputs. The FID architecture can be described as follows: Firstly, the lengthy passage $p$ is divided into multiple segments denoted as $\{seg_1, seg_2, ..., seg_n\}$. Next, $q$ is combined with each segment and encoded separately in the encoder module to generate multiple vector representations $\{e_1, e_2, ..., e_n\}$. Finally, all the vector representations are concatenated as $e_1 \oplus e_2 ... \oplus e_n$ and forwarded to the decoder module for further processing.

## 3.2 Training Process

For the training process, we introduce our training objective, the Sentence Mark (SM) strategy, and the Reasoning Augment Learning (RAL) strategy.

**Training Objective.** Our SADM framework is designed to achieve two training objectives:
- Objective $O_{rationale}$: Training the model to generate a task-related rationale at the prompt of $T_{rationale}$ template.
- Objective $O_{decision}$: Training the model to make a decision based on the generated rationale at the prompt of $T_{answer}$ template.

As shown in Fig. 3, for the objective $O_{rationale}$, we provide the model with $T_{rationale}$, $q$, and $p$ as input, while using the human-annotated rationale $r$ as the learning objective. For objec-

tive $O_{decision}$, we provide $T_{answer}$, $q$, and human-annotated rationale $r$ as input, with the golden target $y$ as the learning objective. We adopt a joint training strategy that involves adding the losses $L_{rationale}$ and $L_{decision}$ of two objectives. Moreover, to calculate the losses $L_{rationale}$ and $L_{decision}$, we employ the teacher-forcing strategy. Given an input sequence $x_1, ..., x_t$ and a target sequence $y_1, ..., y_u$, we maximize the probability $p(y_i|x_1, ..., x_t, y_1, ..., y_{i-2}, y_{i-1})$ for each $y_i$ in the target sequence $y_1, ..., y_u$ to obtain the loss.

**Sentence Mark (SM).** Recent research (Tam et al., 2022) has highlighted the creative nature of generative models. However, in our specific task, there is a concern that the generative model may produce random and irrelevant natural language as a rationale. To mitigate this issue, we adopt the sentence mark strategy (Lakhotia et al., 2020), which involves adding an index number before each sentence in the passage $p$. For instance, a passage consists of $n$ sentences $(s_1, ..., s_n)$. After adding the index, the passage takes the form of $(S_1 : s_1, ..., S_N : s_n)$, where uppercase characters are the sentence indexes. If applying the SM strategy, when optimizing the objective $O_{rationale}$ during the training process, we need to take the sentence indexes of human-annotated rationales as the learning objective instead of the rationale in natural language form.

**Reasoning Augment Learning (RAL).** Cognitive science (Payne et al., 1988) shows that humans have the ability to make reasonable decisions regardless of whether they possess fine-grained information (such as human-annotated rationale) or coarse-grained information (such as the whole passage). Therefore, we imitate human beings, intending to equip the model with the ability to perceive information at different levels of granularity, thereby enhancing its reasoning ability. To accomplish this, we leverage the wealth of information available in supervised data. As shown in Fig. 3, for the objective $O_{decision}$, we add two new formats of training samples. We respectively provide the model with $T_{answer}$, $q$, and $p$ as input, as well as $T_{answer}$, $q$, $r$, and $p$ as input. The golden target $y$ is regarded as the learning objective and the calculated loss is added to $L_{decision}$.

## 3.3 Inference Process

The inference process initiates an auto-regressive process. During the inference process, we em-

ploy a two-stage process as shown in Fig. 2: self-attribution and decision-making.

**Self-attribution.** When the model is prompted with the $T_{rationale}$ template, it generates the rationale based on the claim $q$ and passage $p$. If with the SM strategy, the model first generates sentence indexes, which are then used to locate the corresponding rationale from the passage $p$. And if without the SM strategy, the model directly generates the rationale in natural language form.

**Decision-making.** When the model is prompted with the $T_{answer}$ template, it makes a decision based on the generated rationale. Moreover, we provide the model with the option to make a decision by considering a combination of the generated rationale and passage.

## 4 Evaluation Metrics

Consistent with prior work (Paranjape et al., 2020; Ghoshal et al., 2022), we assess task performance using accuracy and evaluate the quality of generated rationales using the Intersection-Over-Union F1 score (IOU F1) and Token F1 (TF1). We also follow the prior research (Wiegreffe et al., 2020) to take Rationale Accuracy (R-Acc) to evaluate the quality of generated rationales. Additionally, we introduce a novel metric, the Reasoning Success Quotient (RSQ), to gauge the extent of the reliable link between the generated rationale and model decision.

**IOU F1 and TF1 metrics.** The IOU F1 and TF1 metrics are used to evaluate the quality of rationale at the sentence level and the token level respectively. Whether at the sentence level or the token level, the precision and recall rates are calculated for each test sample. Based on them, the F1 value can be calculated. To obtain the IOU F1 or TF1 metrics, the corresponding F1 values of all test samples are averaged. The detailed calculation process is described in App. A.

**R-Acc metric.** We first train a model $f$ with questions and annotated rationales and then evaluate the performance (accuracy) of model $f$ on generated rationales to reflect the quality of the generated rationales.

**RSQ metric.** We propose the RSQ metric to measure the reliable link between the generated rationale and model decision. Specifically, we categorize the test samples into four classes:

- $r_c d_c$: Samples where both the generated rationale and model decision are correct.
- $r_w d_w$: Samples where both the generated rationale and model decision are wrong.
- $r_c d_w$: Samples where the generated rationale is correct, but the model decision is wrong.
- $r_w d_c$: Samples where the generated rationale is wrong, but the model decision is correct.

The RSQ metric is calculated as follows.

$$RSQ = \frac{Num(r_c d_c + r_w d_w)}{Num(r_c d_c + r_w d_w + r_c d_w + r_w d_c)} \tag{1}$$

where $Num$ represents the number of samples. As for how to assess whether the generated rationale or model decision is correct, for the model decision, we determine whether the predicted target aligns with the golden target. For the generated rationale, we assess its correctness using the recall (described in detail in App. A). If the recall rate exceeds a certain threshold, mainly set at 0.5 in our work, we consider the generated rationale to be correct.

Based on the RSQ metric, we also propose RSQ-W and RSQ-C metrics to guide a more detailed analysis. The RSQ-W measures the proportion of wrong decisions made by the model when the model attributes the correct rationales. The RSQ-W metric is as follows:

$$RSQ-W = \frac{Num(r_c d_w)}{Num(r_c d_c + r_c d_w)} \tag{2}$$

The RSQ-C measures the proportion of correct decisions made by the model when the model attributes the wrong rationales. The RSQ-C metric is as follows:

$$RSQ-C = \frac{Num(r_w d_c)}{Num(r_w d_w + r_w d_c)} \tag{3}$$

## 5 Experiment

### 5.1 Datasets

The statistics of datasets used in our experiments are shown in Tab. 2. The FEVER (Thorne et al., 2018) dataset aims to judge whether the given passage supports or refutes the claim. The MultiRC (Khashabi et al., 2018) dataset aims to assign True or False to the question concatenating with the answer choice based on the given passage. The BoolQ (Clark et al., 2019) dataset aims to answer the question with True or False labels based on the given passage. The Evidence Inference (Evi

| | Perf.↑ | IOU F1↑ | TF1↑ | R-Acc↑ | RSQ↑ | RSQ-W↓ | RSQ-C↓ |
|---|---|---|---|---|---|---|---|
| **FEVER** | | | | | | | |
| BERT2BERT | 85.0 | 81.7 | - | - | - | - | - |
| WT5 | 91.9 | 76.6 | 85.3 | - | 74.9 | 6.7 | 86.9 |
| WT5-INV | 91.4 | 76.5 | 84.9 | - | 75.4 | 6.8 | 85.1 |
| FID-Ex($C$=1) | 92.7 | 85.4 | 86.4 | 92.1 | 82.2 | 5.5 | 82.7 |
| SADM($C$=1) | **93.1** | **85.9** | **86.9** | **92.2** | **83.5** | **4.9** | **81.6** |
| **MultiRC** | | | | | | | |
| BERT2BERT | 63.3 | 41.6 | - | - | - | - | - |
| WT5 | 78.0 | 68.0 | 76.6 | - | 69.6 | 20.7 | 72.3 |
| WT5-INV | 77.2 | 66.6 | 75.5 | - | 68.4 | 21.7 | 72.5 |
| FID-Ex($C$=1) | 79.1 | 72.0 | 77.4 | 77.6 | 72.1 | 19.8 | 72.6 |
| SADM($C$=1) | **80.1** | **72.9** | **78.1** | **78.8** | **75.2** | **18.4** | **69.3** |
| **BoolQ** | | | | | | | |
| BERT2BERT | 62.3 | 31.5 | - | - | - | - | |
| WT5 | 71.8 | 44.1 | 63.2 | - | 54.0 | 23.6 | 67.4 |
| WT5-INV | 69.7 | 42.6 | 61.6 | - | 52.8 | 26.3 | 66.1 |
| FID-Ex($C$=1) | 73.9 | 52.3 | 64.2 | 72.8 | 61.9 | 20.1 | 65.5 |
| FID-Ex($C$=10) | 72.4 | 52.2 | 64.3 | 73.2 | 61.5 | 22.8 | 64.9 |
| SADM($C$=1) | **74.3** | **52.9** | **64.5** | 72.9 | **63.1** | **17.1** | **62.9** |
| SADM($C$=10) | 72.5 | 51.9 | 64.3 | **73.6** | 61.4 | 22.6 | 63.9 |
| **Evi Inf** | | | | | | | |
| BERT2BERT | 70.8 | **53.9** | - | - | - | - | - |
| WT5 | 66.8 | 21.1 | 55.2 | - | 46.6 | 17.9 | 62.8 |
| WT5-INV | 62.9 | 22.4 | 55.7 | - | 52.1 | 15.9 | 57.9 |
| FID-Ex($C$=1) | 63.5 | 32.6 | 51.3 | 66.3 | 54.0 | 23.1 | 57.7 |
| FID-Ex($C$=10) | 75.4 | 51.4 | 66.4 | 75.2 | 66.1 | 12.9 | 60.5 |
| SADM($C$=1) | 68.3 | 30.9 | 49.1 | 65.4 | 57.5 | **8.4** | **57.4** |
| SADM($C$=10) | **75.6** | 52.2 | **67.9** | **76.0** | **68.0** | 11.4 | 58.7 |
| **Mov Rev** | | | | | | | |
| BERT2BERT | 86.0 | 15.7 | - | - | - | - | - |
| WT5 | 90.9 | 30.2 | 50.9 | - | 21.1 | 8.7 | 90.9 |
| WT5-INV | - | - | - | - | - | - | - |
| FID-Ex($C$=1) | 90.5 | 57.1 | 68.2 | 87.9 | 68.8 | 7.8 | 86.4 |
| FID-Ex($C$=6) | 96.0 | 57.8 | 67.3 | 95.5 | 41.7 | 2.5 | 95.8 |
| SADM($C$=1) | 94.9 | **63.9** | **73.2** | 89.9 | **90.5** | 4.3 | **84.6** |
| SADM($C$=6) | **96.5** | 62.7 | 71.4 | **96.5** | 55.8 | **1.8** | 94.5 |

Table 1: Experimental results(%) in full-supervised scenarios. Perf. represents task performance and $C$ represents the count of segments set in FID architecture.

Inf) (Lehman et al., 2019) dataset concatenates the (intervention, outcome, comparator) triplet into the question, and aims to judge whether the intervention significantly increases, decreases, or has no effect on the outcome based on the given passage. The Movie Reviews (Mov Rev) (Zaidan and Eisner, 2008) dataset aims to analyze the sentiment of the given passage with positive or negative labels, where the question is uniformly set to "What is the sentiment of this review?". Overall, all five datasets belong to the reasoning tasks. Moreover, the ERASER benchmark provides the annotated rationale at the phrase level for the Mov Rev dataset, and the sentence level for the others. Following the prior work, we convert the phrase level to the sentence level annotations.

| | Train / Dev / Test | # Toks | # Sents |
|---|---|---|---|
| FEVER | 97,957 / 6,122 / 61,111 | 288 | 11 |
| MultiRC | 24,029 / 3,214 / 4,848 | 300 | 14 |
| BoolQ | 6,363 / 1,491 / 2,807 | 3,391 | 165 |
| Evi Inf | 7,958 / 972 / 959 | 4,658 | 153 |
| Mov Rev | 1,600 / 200 / 200 | 774 | 37 |

Table 2: Statistics of datasets in our experiments.

## 5.2 Training Details

Consistent with previous work (Paranjape et al., 2020; Ghoshal et al., 2022), we select T5-base as our main model and use the integrated interface T5ForConditionalGeneration[1] from huggingface to load the model. We run all experiments on a single NVIDIA 80g-a100 GPU machine. We set the learning rate to 1e-4, the batch size to 16, and the total

---
[1]https://huggingface.co/docs/transformers/.

training steps to 15,000 steps, where we evaluate the IOU F1 metric on the validation set every 500 steps to choose the best checkpoint. For datasets with lengthy input, we apply the FID architecture. For the BoolQ, Mov Rev, and Evi Inf datasets, we use a maximum of 512 subword tokens per segment input, where we use 10 segments for the BoolQ and Evi Inf datasets and 6 for the Mov Rev dataset. For the FEVER and MultiRC datasets, we only use one segment and set the maximum input subword lengths to 512 and 1024 respectively.

## 5.3 Baselines

In our study, we compare our SADM framework against several baselines, including BERT2BERT, IB, WT5, WT5-INV, and FID-Ex frameworks. The experimental results for BERT2BERT and IB are reported from the original work, where BERT2BERT is applied in full-supervised scenarios and IB is applied in semi-supervised scenarios. For WT5, WT5-INV, and FID-Ex frameworks, we re-implemented them based on the details provided in the original work. However, it is important to note that there is limited availability of open-source code for these baselines, which presents a challenge in aligning the TF1 metric. To ensure fairness, we do not report the TF1 metric mentioned in the prior work. On the other hand, for task performance and IOU F1 metrics, we successfully aligned them.

## 5.4 Experiment Results

**Full-supervised scenario.** We select the WT5, WT5-INV,FID-Ex and BERT2BERT frameworks as baselines. Since the WT5-INV framework can not obtain a stable performance on the Mov Rev dataset, we do not report its results. As shown in Tab. 1, experimental results demonstrate the promising potential of the SADM framework. For both task performance (Perf.) and the quality of rationale (IOU F1, TF1, and R-Acc), our framework demonstrates varying degrees of improvement across five datasets. Notably, our framework has exhibited more significant improvements in the RSQ metric, which indicates a more reliable link between the generated rationale and model decision. Specifically, we observe 1.3 points improvement on the FEVER dataset, 3.1 points improvement on the MultiRC dataset, 1.2 points improvement on the BoolQ dataset, 1.9 points improvement on the Evi Inf dataset, and 21.7 points improvement on the Mov Rev dataset. Furthermore, experimental results show that the FID architecture only pro-

| | Perf. | IOU F1 | TF1 | RSQ |
|---|---|---|---|---|
| **FEVER** | | | | |
| IB | 88.8 | 66.6 | - | - |
| FID-Ex($C$=1) | 91.5 | 83.9 | 85.3 | 82.1 |
| SADM ($C$=1) | **92.1** | **84.8** | **86.2** | 82.8 |
| **MultiRC** | | | | |
| IB | 66.4 | 54.4 | - | - |
| FID-Ex($C$=1) | 78.4 | 71.5 | 76.8 | 72.3 |
| SADM($C$=1) | **79.9** | **72.6** | **77.6** | **74.4** |
| **BoolQ** | | | | |
| IB | 63.4 | 32.3 | - | - |
| FID-Ex($C$=1) | 70.7 | 46.4 | 60.6 | 55.8 |
| FID-Ex($C$=10) | 65.7 | 49.5 | 60.9 | 53.1 |
| SADM($C$=1) | **73.9** | **50.3** | **61.5** | **61.7** |
| SADM($C$=10) | 73.4 | 47.5 | 61.3 | 57.8 |
| **Evi Inf** | | | | |
| IB | 46.7 | 13.3 | - | - |
| FID-Ex($C$=1) | 55.1 | 26.2 | 47.8 | 57.6 |
| FID-Ex($C$=10) | 64.1 | 43.1 | 61.3 | 57.7 |
| SADM($C$=1) | 59.2 | 23.9 | 41.2 | 59.7 |
| SADM($C$=10) | **73.6** | **45.6** | **62.9** | **65.8** |
| **Mov Rev** | | | | |
| IB | 85.4 | 43.4 | - | - |
| FID-Ex($C$=1) | 86.4 | **59.4** | **70.4** | 73.4 |
| FID-Ex($C$=6) | **94.4** | 55.6 | 66.5 | 46.7 |
| SADM($C$=1) | 87.9 | 56.9 | 68.9 | **76.3** |
| SADM($C$=6) | 91.1 | 54.9 | 67.3 | 50.2 |

Table 3: Experimental results(%) in semi-supervised scenarios. Perf. represents task performance and $C$ represents the count of segments set in FID architecture.

vides improvements for the Evi Inf dataset, whereas it does not exhibit substantial gains for the BoolQ and Evi Inf datasets. We attribute this observation to the fact that, in the Evi Inf dataset, rationale tends to appear in the middle or toward the end of the passage. Hence, addressing the limitation of input length with the FID becomes imperative.

**Semi-supervised scenario.** Considering the expensive cost of rationale annotation, semi-supervised scenarios are more likely to be applied in the real world. We select the IB framework, a variant of the BERT2BERT framework, and the FID-Ex framework, which demonstrates good performance in the full-supervised scenario, as baselines. Following previous settings, we utilize only 25% of the training data with annotated rationales. As shown in Tab. 3, on the Mov Rev dataset, our SADM framework achieves lower performance than the FID-Ex framework in task performance and the quality of rationale but still outperforms in RSQ metric. On the other four datasets, we observe an average improvement of 3.7 points in task performance, 1.3 points in IOU F1, 0.9 points in TF1, and 4.2 points in the RSQ metric. Over-

| | Perf. | IOU F1 | TF1 | RSQ | RCP |
|---|---|---|---|---|---|
| **FEVER** | | | | | |
| w/o SM | 92.4 | 76.3 | 85.1 | 75.9 | - |
| w/o RAL | 92.0 | 85.6 | 86.6 | 82.5 | 93.7 |
| SADM | 93.1 | **85.9** | **86.9** | 83.5 | **94.5** |
| w passage | **93.3** | - | - | 76.3 | - |
| **MultiRC** | | | | | |
| w/o SM | 77.9 | 67.1 | 75.3 | 70.6 | - |
| w/o RAL | 79.5 | 72.8 | 77.8 | 74.1 | 79.9 |
| SADM | 80.1 | **72.9** | **78.1** | 75.2 | **81.5** |
| w passage | **80.9** | - | - | 74.8 | - |
| **BoolQ** | | | | | |
| w/o SM | 72.8 | 44.6 | 63.8 | 57.4 | - |
| w/o RAL | 72.9 | 52.1 | 64.0 | 62.4 | 80.3 |
| SADM | 74.3 | **52.9** | **64.5** | 63.1 | **80.7** |
| w passage | **75.2** | - | - | 61.3 | - |
| **Evi Inf** | | | | | |
| w/o SM | 67.4 | 21.2 | 55.2 | 47.6 | - |
| w/o RAL | 63.8 | 30.9 | 49.1 | 56.9 | 88.4 |
| SADM | **68.3** | **31.2** | **49.9** | 57.5 | **91.4** |
| w passage | 66.9 | - | - | 57.2 | - |
| **Mov Rev** | | | | | |
| w/o SM | 89.5 | 30.5 | 49.3 | 23.1 | - |
| w/o RAL | 89.4 | 56.7 | 68.6 | 66.8 | 99.4 |
| SADM | 94.9 | **63.9** | **73.2** | 90.5 | **99.9** |
| w passage | **95.9** | - | - | 89.4 | - |

Table 4: Experimental results(%) of our ablation study.

| Threshold | 0.6 | 0.7 | 0.8 | 0.9 | 1.0 |
|---|---|---|---|---|---|
| **FEVER** | | | | | |
| FID-Ex($C$=1) | 75.4 | 75.2 | 75.2 | 75.1 | 75.1 |
| SADM($C$=1) | **77.2** | 76.8 | 76.7 | 76.7 | 76.7 |
| **MultiRC** | | | | | |
| FID-Ex($C$=1) | 59.9 | 53.8 | 52.9 | 52.9 | 52.9 |
| SADM($C$=1) | **63.9** | 56.9 | 55.8 | 55.8 | 55.8 |
| **BoolQ** | | | | | |
| FID-Ex($C$=1) | 59.7 | 57.1 | 55.6 | 52.6 | 52.4 |
| FID-Ex($C$=6) | 59.3 | 57.7 | 55.8 | 55.8 | 52.7 |
| SADM($C$=1) | **61.1** | 58.8 | 57.4 | 54.9 | 54.8 |
| SADM($C$=6) | 59.2 | 56.9 | 54.8 | 52.4 | 52.2 |
| **Evi Inf** | | | | | |
| FID-Ex($C$=1) | 46.7 | 46.6 | 46.6 | 46.6 | 46.6 |
| FID-Ex($C$=6) | 54.5 | 52.5 | 52.5 | 52.5 | 52.5 |
| SADM($C$=1) | 47.2 | 47.1 | 47.1 | 47.1 | 47.1 |
| SADM($C$=6) | **56.9** | 55.1 | 54.9 | 54.9 | 54.9 |
| **Mov Rev** | | | | | |
| FID-Ex($C$=1) | 56.8 | 41.7 | 27.1 | 17.6 | 13.1 |
| FID-Ex($C$=6) | 41.7 | 25.6 | 15.6 | 6.5 | 4.5 |
| SADM($C$=1) | **84.9** | 72.9 | 50.3 | 23.6 | 14.6 |
| SADM($C$=6) | 55.8 | 35.7 | 21.1 | 8.5 | 6.1 |

Table 5: Experimental results(%) with different thresholds chosen in the RSQ metric.

all, our framework demonstrates more significant advantages in the semi-supervised scenario.

# 6 Analysis

In our analysis, we conduct ablation experiments (in Sec. 6.1) to evaluate the effectiveness of each strategy in our SADM framework. We also consider different choices of threshold in the RSQ metric (in Sec. 6.2) to provide robust results. Furthermore, we provide quantitative analysis to evaluate the lack of a reliable link between the rationale and model decision in the competitive FID-Ex framework, which will be shown in detail in App. B.

## 6.1 Ablation Study

We evaluate the performance of SADM without the SM strategy and SADM without the RAL strategy. As shown in Tab. 4, experimental results show that the performance of the SADM framework decreases to a certain extent when either the SM strategy or RAL strategy is removed, which indicates that both the SM strategy and RAL strategy play a positive role. Notably, we specifically verify the effect of the RAL strategy on model reasoning ability. We propose the Rationale-Centric Precision (RCP) metric, which focuses on the pro-

portion of correct decisions that can be made when the model is provided with the annotated rationale at the decision-making stage. Our experimental results show that when the RAL strategy is removed, the RCP metric decreases by an average of 1.3 points across the five datasets. Such a phenomenon underscores the critical significance of the RAL strategy in enhancing the model's reasoning ability at the decision-making stage.

Additionally, as introduced in Sec. 3.1, the passage is an optional input at the decision-making stage. For our SADM framework, the model can make a decision based on a combination of generated rationale and passage. As shown in Tab. 4, experimental results show that considering both generated rationale and passage at the decision-making stage can improve task performance, but slightly damage the reliable link between the generated rationale and model decision. Such a phenomenon is reasonable. When the model is provided with more contextual information, it will make more correct decisions, but at the same time, the information it focuses on will be more scattered.

## 6.2 Choice of Threshold

To ensure a standardized evaluation of the generated rationale, we have selected a threshold of 0.5 for the recall rate in the RSQ metric. However, we are concerned that the choice of the threshold will

bring bias in our experimental results. Therefore, we have also conducted experiments with alternative threshold values of 0.6, 0.7, 0.8, 0.9, and 1.0, which allows us to present a robust evaluation. As shown in Tab. 5, compared to strong baselines, results consistently showcase the significant advantage of our SADM framework across the entire range of threshold values in the RSQ metric.

## 7 Conclusion

Our proposed SADM framework establishes a more reliable link between the generated rationale and model decision while improving task performance and rationale quality. Furthermore, we observe significant advantages in semi-supervised scenarios. In the future, we will explore how to optimize our framework to attain greater performance improvements.

## 8 Limitation

Despite the numerous advantages exhibited by our SADM framework, we believe that it still has the following limitations:
- Our SADM framework achieves good performance in full-supervised and semi-supervised scenarios, but in the future, we will put more effort into thinking about how to apply our SADM framework in the unsupervised scenario.
- In our work, we only design natural language oriented toward human understanding as prompt templates. Is this necessarily the best for the model? We will further explore their influence.

## 9 Ethics Statement

We conduct all experiments on publicly available datasets with authorization from the respective maintainers. All data we use do not involve personal privacy, ethical issues, or sensitive topics.

## Acknowledgements

This work was supported by the National Key R&D Program of China [2021ZD0113302]; the National Natural Science Foundation of China [62206079]; and the Heilongjiang Provincial Natural Science Foundation of China [YQ2022F006].

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

## A IOU F1 and TF1

**IOU F1.** IOU F1 is used to assess the quality of rationale at the sentence level. As shown in the left of Fig. 4, it is assumed that the annotated rationale consists of four sentences (highlighted in green) and the generated rationale consists of three sentences (highlighted in yellow), of which two sentences match the sentences from annotated rationales (connected part). In this sample, the precision ratio is calculated as $2/3 = 0.67$, which represents the ratio of matched sentences to the total number of sentences from the generated rationale. The recall ratio is calculated as $2/4 = 0.5$, which represents the ratio of matched sentences to the total number of sentences from the annotated rationale. By considering both precision and recall, the F1 value can be calculated for each test sample. Finally, the F1 values of all test samples are averaged to obtain the Macro IOU F1 score.

Additionally, as shown in the right of Fig. 4, it illustrates the approach for determining whether two sentences match. A ratio is computed by dividing the length of the intersection between two sentences by the length of their union. If this ratio surpasses a specific threshold (typically set as 0.5 in prior work), the two sentences are considered to be a match. In our study, we utilize the longest common substring to determine the length of the sentence intersection, and the matching score is subsequently calculated.

**TF1.** TF1 is used to assess the quality of rationale at the token level. In each test sample, we define the set of tokens from the annotated rationale as $Q_1$, and the set of tokens from the generated rationale as $Q_2$. The intersection of these two sets is denoted as $Q$. The precision rate is computed by dividing the length of set $Q$ by the length of set $Q_2$, while the recall rate is calculated by dividing the length of set $Q$ by the length of set $Q_1$. By considering precision and recall, the F1 value can be calculated. Finally, the F1 values of all test samples are averaged to obtain the Macro TF1 score.

## B Quantitative Analysis

Our competitive baseline FID-Ex framework involves generating the classification decision followed by rationale in a parallel way. To quantitatively assess the reliable link between the generated rationale and model decision, we conduct experiments illustrated in Fig. 5. We apply a masking

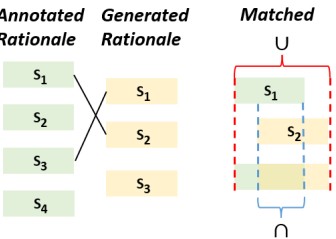

Figure 4: Illustration of IOU F1 metric.

---

*Claim*: **Nestor Carbonell played Godzilla in Lost.**

*Passage*: *He is perhaps most famous for his roles as Richard Alpert in ABC drama series Lost.* **... He is also known for his regular roles as Luis Rivera on the sitcom Suddenly Susan , and Batmanuel on the live-action sitcom The Tick. ...**

*Gold Answer*: **REFUTE.**

> **Origin**

*Model Output*: **Answer: REFUTE. Rationale: He is perhaps most famous for his roles as Richard Alpert in ABC drama series Lost.**

> **Mask**

*Model Output*: **Answer: <PAD>. Rationale: He is perhaps most famous for his roles as Richard Alpert in ABC drama series Lost.**

---

Figure 5: Illustration of the method for analyzing the FID-Ex framework.

technique to the model decision and then initialize the model with the prompt "$Answer :< pad > Explanation :$" to encourage the generation of rationale. We believe that if there is a reliable link between the generated rationale and the model decision, the IOU F1 and TF1 metrics will change significantly after the masking of the model decision. However, interestingly, as presented in Tab. 6, despite masking the model decision, we observe minimal changes in both the IOU F1 and TF1 metrics. This somewhat suggests that, for the FID-Ex framework, the rationale may be generated independently, with no reliable link to the model decision.

| FID-Ex | IOU F1 | | TF1 | |
|---|---|---|---|---|
| | Origin | Mask | Origin | Mask |
| FEVER | 85.4 | 85.2 | 86.4 | 86.3 |
| MultiRC | 72.1 | 72.2 | 77.4 | 77.3 |
| BoolQ | 52.3 | 49.1 | 64.2 | 59.8 |
| Evi Inf | 32.6 | 32.3 | 51.3 | 51.2 |
| Mov Rev | 57.1 | 57.3 | 68.2 | 68.1 |

Table 6: Quantitative analysis of FID-Ex framework.