# OpenReview forum: "Make Your Decision Convincing! A Unified Two-Stage Framework: Self-Attribution and Decision-Making"
_EMNLP/2023/Conference — EMNLP 2023 Findings_

### Official Review · Reviewer_TZrn · 2023-08-04

**Soundness:** 3

**Excitement:**

3: Ambivalent: It has merits (e.g., it reports state-of-the-art results, the idea is nice), but there are key weaknesses (e.g., it describes incremental work), and it can significantly benefit from another round of revision. However, I won't object to accepting it if my co-reviewers champion it.

**Paper Topic And Main Contributions:**

This paper finds that the problem that the rationale does not genuinely support and convincingly justify the model decision. In specific, the model may make correct decisions while attributing wrong rationales or make poor decisions while attributing correct rationales. To solve such problems, this paper proposes self-attribution and decision-making (SADM), which jointly optimize both the self-attribution and decision-making objectives. SADM is prompted to extract rationales and employs a reasoning augment learning strategy to decision-making. Extensive experimental results show the effectiveness of the proposed method.

**Questions For The Authors:**

1. What is the proportion of the unreliable link problem? And how many problems does SADM address?

**Reasons To Accept:**

1. This paper finds an important problem: the link between the rationale and classification decisions generated by models is not reliable.
2. To solve this problem, this paper proposes SADM,  which could jointly optimize both the self-attribution and decision-making objectives. SADM is prompted to extract rationales and employs a reasoning augment learning strategy to decision-making. Extensive experimental results show the effectiveness of the proposed method. Besides, SADM also incorporates Fusion-In-Decoder to address the challenges of lengthy texts and Sentence Mark strategy to mitigate the issue of irrelevant rationale generation.
3. Extensive experiments show the effectiveness of SADM. The experiments include full-supervised scenarios and semi-supervised scenarios, and SADM both achieve the best performance compared to the baselines.

**Reasons To Reject:**

1. The proposed problem includes two aspects: a. model may make correct decisions while attributing wrong rationales; b. make poor decisions while attributing correct rationales. Actually, the first problem seems to be normal. Considering the faithfulness of rationale extraction,  the model may just make correct decisions while attributing wrong rationales. This is a long-term research topic in recent interpretability community. Therefore, only the second problem need to solve.
2. This paper only qualitatively analyze the problem (Examples in Figure 1). However, this is not enough. To make this paper more qualitative, this paper should statistically analyze the proportion of the unreliable link problem. And this paper should show whether the proposed method could solve them. This would make this paper more complete.

**Reproducibility:**

3: Could reproduce the results with some difficulty. The settings of parameters are underspecified or subjectively determined; the training/evaluation data are not widely available.

**Reviewer Confidence:**

3: Pretty sure, but there's a chance I missed something. Although I have a good feel for this area in general, I did not carefully check the paper's details, e.g., the math, experimental design, or novelty.

---

> ### Author Rebuttal · Authors · 2023-08-29
>
> ## Response to Reviewer TZrn
> Thanks for taking the time to review our work. Thank you for recognizing our efforts in addressing a focused and meaningful problem, and appreciating the effectiveness of our proposed method and the adequacy of our experiments. We have summarized your main questions as follows and will address them one by one.
>
> ### Q1: What is the proportion of the unreliable link problem? And how many problems does SADM address?
>
> In our work, we introduce the RSQ metric to measure the correlation between generated rationales and decisions. However, for two cases: a. the model might make the correct decision but provide incorrect rationales; b. make incorrect decisions while giving the correct rationales. The RSQ metric fails to indicate whether our proposed framework can improve the above cases separately and to what extent. Therefore, to make our paper more complete, we provide another qualitative analysis as follows.
>
> As introduced in Lines 336-343, we divide the test samples into four classes:
> - $r_c d_c$: Samples where both the generated rationale and model decision are correct.
> - $r_w d_w$: Samples where both the generated rationale and model decision are wrong.
> - $r_c d_w$: Samples where the generated rationale is correct, but the model decision is wrong.
> - $r_w d_c$: Samples where the generated rationale is wrong, but the model decision is correct.
>
> For the case that the model may make wrong decisions while attributing correct rationales, we propose the RSQ-W metric. The purpose is to observe the proportion of wrong decisions made by the model when the model attributes the correct rationales. The RSQ-W metric is as follows:
> $$
> RSQ-W=\frac{Num(r_c d_w)}{Num(r_c d_c + r_c d_w)}
> $$
>
> For the case that the model may make correct decisions while attributing wrong rationales, we propose the RSQ-C metric. The purpose is to observe the proportion of correct decisions made by the model when the model attributes the wrong rationales. The RSQ-C metric is as follows:
> $$
> RSQ-C=\frac{Num(r_w d_c)}{Num(r_w d_w + r_w d_c)}
> $$
>
> The experimental results are as follows.
>
>
> |       | RSQ-W | RSQ-C |
> |-----------------|-----------------|-----------------|
> |  FEVER  |     |      |
> |  WT5  |  6.67%   | 86.94%    |
> | WT5-INV| 6.76%   | 85.06%    |
> |FID-Ex(C=1)| 5.47% | 82.72%   |
> |SADM(C=1)|  **4.97%**  | **81.57%**   |
> |  MultiRC  |     |      |
> |  WT5  |  20.68%   | 72.34%    |
> | WT5-INV| 21.71%   | 72.52%    |
> |FID-Ex(C=1)| 19.81% | 72.54%   |
> |SADM(C=1)|  **18.36%**  | **69.34%**   |
> |  BoolQ  |     |      |
> |  WT5  |  23.56%   | 67.41%    |
> | WT5-INV| 26.26%   | 66.10%    |
> |FID-Ex(C=1)| 20.05% | 65.53%   |
> |FID-Ex(C=10)|  22.80%  | 64.96%   |
> |SADM(C=1)| **17.01%** | **62.95%**   |
> |SADM(C=10)|  22.61%  | 63.89%  |
> |  Evi Inf  |     |      |
> |  WT5  |  17.91%   | 62.80%    |
> | WT5-INV| 15.96%   | 57.97%    |
> |FID-Ex(C=1)| 23.08% | 57.73%   |
> |FID-Ex(C=10)|  12.87%  | 60.52%   |
> |SADM(C=1)| **8.42%** | **57.45%**   |
> |SADM(C=10)|  11.44%  | 58.75%  |
> |  Mov Rev  |     |      |
> |  WT5  |  8.71%   |    90.95% |
> | WT5-INV| -   | -    |
> |FID-Ex(C=1)| 7.86% |  86.44%  |
> |FID-Ex(C=10)| 2.50%   | 95.80%  |
> |SADM(C=1)|   4.30%   |   **84.62%** |
> |SADM(C=10)|  **1.85%**  |  94.51% |
>
> In this evaluation, a smaller experimental value corresponds to a better performance of the model. Experimental results show that our proposed SADM framework achieves stable improvements for both cases. I hope the quantitative analysis we provide will address your concerns.
>
> Overall, thanks again for taking the time to review our work. I hope our response can clear up your confusion and improve your perception of our work, which is of the utmost importance to us. In the camera-ready version, we will add the above additional analysis to our paper to make our work more complete.

---

### Official Review · Reviewer_WBPQ · 2023-08-05

**Soundness:** 3

**Excitement:**

2: Mediocre: This paper makes marginal contributions (vs non-contemporaneous work), so I would rather not see it in the conference.

**Missing References:**

[1] Wiegreffe et al., Measuring Association Between Labels and Free-Text Rationales

[2] Jain et al., Learning to Faithfully Rationalize by Construction

**Paper Topic And Main Contributions:**

This paper focuses on the research problem of loosely linked model-generated rationales to its predictions, such as correct predictions associated with wrong rationales or wrong predictions associated with correct rationales. To address this problem, a two-stage framework called Self-Attribution and Decision-Making (SADM) is proposed, where the model is trained jointly with both the self-attribution and decision-making objectives during training, while first prompted to extract the rationale from the given input and then prompted to utilize the rationale to make a decision during inference. Besides, the sentence mark (SM) strategy is applied to prevent the generative model from generating random and irrelevant rationale. And reasoning augment learning (RAL) is used to enhance the model’s reasoning ability. The proposed method is compared with several baselines (BERT2BERT, IB, WT5, WT5-INV, and FID-Ex) on five datasets in the ERASER benchmark. Experiments demonstrate the effectiveness of the proposed method.

**Reasons To Accept:**

The proposed method seems effective as compared to baseline methods.

**Reasons To Reject:**

- The distinction between this work and previous pipeline frameworks [1, 2] remains unclear, despite the application of new techniques such as SM and RAL. Additionally, the omission of important baseline methods in this paper further complicates the evaluation. Without a clear comparison and distinction, the novelty of this work remains ambiguous. To strengthen the paper's contribution, it is crucial to explicitly highlight the unique aspects that set it apart from existing approaches and conduct a comprehensive comparison with relevant baseline methods. This will provide a clearer understanding of the original contributions and potential advancements made by this work.
- The method section (Sec. 3) could be improved to enhance its clarity. It would greatly benefit from including background information on previous techniques (e.g., SM, RAL) to provide readers with a better context for understanding the proposed method. Additionally, explicitly formulating the objectives (Sec. 3.2) would make the proposed method more accessible and understandable.
- Lack of citations in Section 2.2 (e.g., BERT2BERT, IB, WT5 and FID-Ex).
- It would be better to include human evaluations on the generated rationales.

[1] Wiegreffe et al., Measuring Association Between Labels and Free-Text Rationales

[2] Jain et al., Learning to Faithfully Rationalize by Construction

**Reproducibility:**

3: Could reproduce the results with some difficulty. The settings of parameters are underspecified or subjectively determined; the training/evaluation data are not widely available.

**Reviewer Confidence:**

4: Quite sure. I tried to check the important points carefully. It's unlikely, though conceivable, that I missed something that should affect my ratings.

---

> ### Author Rebuttal · Authors · 2023-08-29
>
> ## Response to Reviewer WBPQ
> Thanks for taking the time to review our work. Thank you for acknowledging the effectiveness of our proposed SADM framework. We have summarized your main questions as follows and will address them one by one.
>
> ### Q1: What are the differences compared to the previous work[1]?
>
> We summarize the differences from the work [1] in the following four aspects:
>
> - **Different scenarios:** The work[1] focuses on the free-text rationale scenario, whereas our work focuses on the extracted rationale scenario.
> - **Different motivations:** The work[1] aims to conduct an analysis to determine whether the pipeline framework (I-R, R-O) or the end-to-end (I-RO) framework is better suited for the free-text rationale scenario. Our work aims to enhance the reliable link between generated rationales and decisions in the extracted rationale scenario. To achieve this, we introduce a novel SADM framework.
> - **Different conclusions:** Due to different scenarios, we come to different conclusions.  The work[1] concludes that pipeline frameworks are not suitable for the free-text rationale scenario, while our work concludes that pipeline frameworks are well suited for the extracted rationale scenario.
> - **Different analysis perspectives:** In the work[1], the analysis of the correlation between generated rationales and decisions focused solely on the end-to-end framework, without comparing it to the pipeline framework. In contrast, our work comprehensively compares the performance of the pipeline framework and the end-to-end framework, including the performance of decisions, the quality of generated rationales, and the correlation between generated rationales and decisions.
>
> To further resolve your confusion, we provide some additional clarification as follows:
> - **Free-text rationale scenario vs extracted rationale scenario.** For the free-text rationale scenario, the goal is to have the model make the correct choice to question based on the candidate choices and generate coherent paragraphs of free text as rationales to support its decision. For the extracted rationale scenario, the goal is to have the model provide the answer to the question based on the given passage while extracting rationales from the passage to support its decision.
>
> - **Why different conclusions are drawn in different scenarios?** The work[1] has revealed that in the free-text rationale scenario, pipeline frameworks face the following challenges: 1. Cascading errors due to poor quality rationale generated during the I-R phase. 2. The insufficient generated rationales result in poor performance. 3. Double the number of parameters to reach comparable performance to an end-to-end (I→OR) framework, but still often perform worse. However, these challenges appear to be non-existent in the extracted rationale scenario. Concerning challenges 1 and 2, as illustrated in Table 2 of our paper, our SADM framework ensures the quality of generated rationales during the I-R phase. We attribute this capability to the feasibility of employing information extraction (IE) tasks on the t5-base model. However, for the free-text rationale scenario, generating free-text still presents challenges. Besides, for the performance of models, our proposed pipeline framework has surpassed the end-to-end framework. As for challenge 3, this is addressed in our proposed framework. As depicted in Lines 247-252 and shown in Figure 2, our SADM framework shares parameters between the two stages during the training phase and employs a joint training strategy. For the inference stage, we use the shared model to perform two stages separately. Experiments show that our framework can achieve better performance without adding additional parameters.
>
> - **The analysis for the pipeline framework was not considered in the work[1].** Although the pipeline framework exhibited poorer performance in the free-text rationale scenario, this does not necessarily imply that the pipeline framework would perform a worse correlation between generated rationales and decisions. For instance, due to wrong rationales, the model provides incorrect decisions, also indicating a strong correlation.
>
>
>
> ### Q2: Missing the evaluation[1].
>
> The work [1], to address the challenge of the free-text rationale scenario, introduces novel evaluation metrics designed to assess the quality of generated rationales and quantify the correlation between generated rationales and decisions. Conversely, for the extracted rationale scenario, there is already a well-established evaluation system. We adhere to the evaluation metrics IOU and TF1 presented by the ERASER Benchmark [3] to evaluate the quality of generated rationales, consistent with prior research [4] [5] [6]. Building upon the IOU metric, we further introduce a metric tailored to enhance the measurement of the correlation between generated rationales and decisions within the extracted rationale scenario.
>
> - **The quality of generated rationale:** Due to the inadequacy of IOU and TF1 in assessing the quality of free-text rationales, the work[1] proposed a novel evaluation metric. This evaluation metric involves first training a model M with questions and annotated rationales, and then evaluating the performance of model M on generated free-text rationales to reflect the quality of the generated rationales. Despite this evaluation metric not having been previously considered in the extracted rationale scenario, in order to enhance the robustness of our work, we incorporate this evaluation metric and conduct an additional experiment as follows.
>
>
> |   Framework    | FEVER | MultiRC | BoolQ | Evi Inf | Mov Rev |
> |-----------------|-----------------|-----------------|-----------------|-----------------|-----------------|
> | Gold    | 94.08    | 80.41    | 78.56    | 90.09    | 100    |
> | FID-Ex(C=1)    | 92.13    | 77.62    | 72.78    | 66.32    | 87.94    |
> |   FID-Ex(C=10)    | -    | -    | 73.24    | 75.23    | 95.48    |
> | SADM(C=1)    | **92.21**    | **78.75**    | 72.92    | 65.38    | 89.95    |
> | SADM(C=10)    | -    | -    | **73.64**    | **76.02**    | **96.48**    |
>
>
>
> We conduct a comparison between our SADM framework and the state-of-the-art FID-Ex framework. In this evaluation, a larger experimental value corresponds to a higher quality of generated rationales. The pinnacle of performance is represented by the ''Gold'' standard. Notably, the experimental results unequivocally demonstrate that our SADM framework continues to surpass the strong baselines, underscoring its exceptional performance and superiority.
>
>
> - **The correlation between generated rationales and decisions:**  In the free-text rationale scenario, assessing the validity of generated rationales is challenging due to the inherent ambiguity of standard rationales. This makes it difficult to directly measure the correlation between the generated rationale (R) and the output (O). Therefore, in the work[1], for the free-text rationale scenario, a novel evaluation approach is proposed. This approach involves perturbing the input (I), observing changes in both the model's performance (O), and the quality of the generated rationales (R), and assessing if they exhibit similar trends. While this evaluation approach indirectly reflects the correlation between R and O, there remains a concern about coincidental alignment in their change trends. In contrast, in the extracted rationale scenario, where standard rationales are well-defined, the use of statistical metrics proposed by the ERASER benchmark enables the assessment of the validity of each rationale. This facilitates a straightforward measurement of the correlation between R and O, as discussed in Lines 332-354. Therefore, in our work, we propose a new metric RSQ for evaluating the correlation between generated rationales and decisions for the extracted rationale scenario, building on the work of ERASER benchmark [3].
>
>
>
>
>
> ### Q3: Missing a baseline [2].
> The work[3] has indicated that the performance of the pipeline framework introduced in work [2] is notably inferior to that of our selected baseline, the BERT2BERT pipeline framework. It is important to note that our work not only delves into the pipeline framework but also encompasses the comprehensive exploration of end-to-end frameworks. Consequently, we choose the top-performing and most representative methods from each framework as our strong baselines. Therefore, for the pipeline framework, the BERT2BERT framework is chosen by us.
>
>
>
> ### Q4: Further clarification of method.
>
> Thank you for your valuable suggestions, we will provide further background information on the SM technique in the camera-ready version. Besides, we need to clarify that the RAL technique is first proposed in our work, and we will further clarify the motivation of the RAL technique and the overall goal of the SADM framework.
>
> ### Q5: Lack of citations.
>
> Thank you for promptly pointing out the issues in our writing. The citations for WT5 and FID-Ex have been marked in Lines 46-47, and we will cite the references for BERT2BERT[3] and IB[4] in the camera-ready version.
>
> ### Q6: Lack of Human evaluation.
>
> For our extracted rationale scenario, automated metrics are a good measure of the quality of the generated rationale. Recent research[3] [4] [5] [6] over the past three years has uniformly employed IOU and TF1 metrics to assess the quality of generated rationales. Therefore, we keep consistent with them for a fair comparison. Besides, in previous work[3], IOU and TFI metrics have been also shown to align with human evaluations. We believe that human evaluation is more necessary and indispensable for the free-text rationale scenario.
>
> Overall, thanks again for taking the time to review our work. I hope our response can clear up your confusion and improve your perception of our work, which is of the utmost importance to us. In the camera-ready version, we will make changes to some of the details you suggested, such as adding missing references, clarifying the motivation of the method, etc. We will do everything to make our work more convincing.
>
>
> [1]Wiegreffe et al., Measuring Association Between Labels and Free-Text Rationales.
>
> [2]Jain et al., Learning to Faithfully Rationalize by Construction.
>
> [3]Jay DeYoung et al., ERASER: A Benchmark to Evaluate Rationalized NLP Models.
>
> [4]Bhargavi Paranjape et al., An Information Bottleneck Approach for Controlling Conciseness in Rationale Extraction.
>
> [5]Asish Ghoshal Srinivasan Iyer et al., QUASER: Question Answering with Scalable Extractive Rationalization.
>
> [6]Kushal Lakhotia et al., FiD-Ex: Improving Sequence-to-Sequence Models for Extractive Rationale Generation.

---

### Official Review · Reviewer_AqvU · 2023-08-08

**Typos Grammar Style And Presentation Improvements:** 1. Line 552
**Soundness:** 3

**Excitement:**

3: Ambivalent: It has merits (e.g., it reports state-of-the-art results, the idea is nice), but there are key weaknesses (e.g., it describes incremental work), and it can significantly benefit from another round of revision. However, I won't object to accepting it if my co-reviewers champion it.

**Missing References:**

1. Hui Liu, et al. Towards Explainable NLP: A Generative Explanation Framework for Text Classification. ACL 2019.

**Paper Topic And Main Contributions:**

Topic:

This work studies an important problem, which is the consistency between the model predictions and the corresponding generated rationales. This work proposes a two-stage framework, which is to prompt the model to generate the rationale first, and then generate the answer based on the rationale. Experiments on different datasets demonstrate the effectiveness of the proposed method.

Contribution:

This work proposes a framework that can generate consistent answers and rationales.


**Reasons To Accept:**

1. This work studies an important problem, which is meaningful.

2. The proposed method is shown to be effective on various datasets.

3. The paper is well-written and easy to follow.

**Reasons To Reject:**

1. The idea of the proposed method is not new, which makes this work less exciting. Previous works have studied the process of I->R and then IR->O where I, R and O means the input, rationale and output. Hence, the contribution of this work seems to be more on the engineering side.

2. Different variants should be compared to more comprehensively evaluate the performance, e.g., I->RO, I-O and then IO->R, etc. These variants can benefit the evaluation and help to understand the contribution of different components in the method. Also, I suggest the authors explore the potential of LLMs on this problem, since some results show that LLMs seem to be pretty good at generating consistent answers and rationales.

**Reproducibility:**

4: Could mostly reproduce the results, but there may be some variation because of sample variance or minor variations in their interpretation of the protocol or method.

**Reviewer Confidence:**

4: Quite sure. I tried to check the important points carefully. It's unlikely, though conceivable, that I missed something that should affect my ratings.

---

> ### Author Rebuttal · Authors · 2023-08-29
>
> ## Response to Reviewer AqvU
> Thanks for taking the time to review our work. Thank you for recognizing our efforts in addressing a focused and meaningful problem, and appreciating the effectiveness of our proposed method and the high quality of our paper. We have summarized your main questions as follows and will address them one by one.
>
> ### Q1: Since previous works have studied the process of pipeline framework (first I->R and then IR->O), this work is less exciting.
>
> Despite extensive discussions surrounding the pipeline framework, recent work [2] claims that a significant challenge still confronts its implementation.
>
> - Under the same scale of parameters, a considerable performance gap can be observed when compared to the end-to-end frameworks (such as I-OR or I-RO frameworks). This phenomenon is particularly evident as shown in the performance metrics outlined in Table 2 of our paper. The pipeline BERT2BERT framework's performance notably lags behind that of the end-to-end WT5, WT5-INV, and FID-Ex frameworks.
>
> Excitingly, our proposed pipeline framework (SADM) outperforms the end-to-end frameworks without adding any additional parameters. Furthermore, compared to different frameworks, our proposed framework can establish a more reliable link between the generated rationales and decisions. We believe that all of the above are crucial for constructing an explanation model.
>
>
> ### Q2: Different variants should be compared to more comprehensively evaluate the performance, e.g., I->RO, I-O, and then IO->R, etc.
>
> In our work, we have conducted a comprehensive and detailed comparison of different variants. However, expression problems in writing may have caused some confusion for you and made you misunderstand our work. To shed light on the selection process for different variants, we provide the following elucidation.
>
> Our work focuses on explanation models capable of making decisions while providing rationales to support their decisions. For the I-O framework, it lacks the capacity to provide rationales to support their decisions. Therefore, we have not taken the I-O framework into consideration.
>
> For explanation models, there are the following frameworks:
> - I-OR
> - I-RO
> - I-O, O-R
> - I-R, IR-O
>
> We choose the top-performing and most representative method from each framework, establishing a robust baseline for our study. Within our baselines, WT5 and FID-Ex belong to the I-OR framework, WT5-INV belongs to the I-RO framework, and BERT2BERT and IB belong to the I-R, IR-O framework.
>
> As for the I-O, O-R framework, it has rarely been considered in recent work due to its often counterintuitive behavior (as introduced in Lines 145-150). Besides, the work[1] has explicitly highlighted the superior performance of the I-R, R-O framework (the BERT2BERT method) in comparison to the I-O, O-R framework (including methods like LIME and Attention Score). Meanwhile as illustrated in Table 2 and Table 3, our proposed framework significantly outperforms the BERT2BERT framework. Consequently, any concerns surrounding the I-O, O-R framework are redundant.
>
>
>
>
> ### Q3: Explore the potential of LLMs.
>
> In the current era, it is crucial to explore the potential of Large Language Models (LLMs). But in our scenario, we have the following concerns:
>
> - Since LLMs are completely black-box (like chatgpt and llama), we are not sure whether they have seen the test data during the training phase. Therefore, it is difficult for us to provide a fair comparison.
>
> - Evaluating LLMs within a standardized framework is intricate. LLMs often generate some free-text rationales regardless of our control. Therefore, IOU and TF1 metrics are unsuitable.
>
> - In our comparison experiments, the parameter size used by all methods is the same, that is, 220M. If compared to LLMs, the benefit of model parameter size is hard to ablate.
>
> Based on the above concerns, we do not compare LLMs in our main experiments. But we agree with your valuable suggestion that we should explore the potential of LLMs. We have the following plans in the camera-ready version:
> - Clarify our concerns about LLMs.
> - Following parameter-efficient fine-tuning technology (such as LoRA), we apply our SADM framework to the base model Llama to explore the potential of LLMs.
>
> Overall, thanks again for taking the time to review our work. I hope our response can clear up your confusion and improve your perception of our work, which is of the utmost importance to us. In the camera-ready version, we will make changes to some of the details you suggested.
>
> [1]Jay DeYoung et al., ERASER: A Benchmark to Evaluate Rationalized NLP Models.
>
> [2]Wiegreffe et al., Measuring Association Between Labels and Free-Text Rationales.

---

### Meta-Review · Area_Chair_hxLR · 2023-09-21

**Recommendation:** 3

**Metareview:**

The paper proposes a method for self-rationalization. The proposed method is an instance of I->R, IR->O, (given input, generate rationale. Given input + rationale, generate output), which is a good design that 1) improves the accuracy of O because it is conditioned on R, 2) increases the consistency between R and O.

While this setup is not new, e.g.[1], the details of how the I->R model and the IR->O model are implemented led to improved results in the 5 datasets of the ERASER benchmark.

During the rebuttal, the authors added new results to quantify the consistency between R and O using their method vs. previous work, and showed greater consistency. This result is an afterthought, so the authors should make sure to add it to the paper.


[1]Wiegreffe et al., Measuring Association Between Labels and Free-Text Rationales.

---

### Decision · Program_Chairs · 2023-10-07

**Decision:**

Accept-Findings

**Comment:**

The paper proposes a method for self-rationalization. The proposed method is an instance of I->R, IR->O, (given input, generate rationale. Given input + rationale, generate output), which is a good design that 1) improves the accuracy of O because it is conditioned on R, 2) increases the consistency between R and O.

While this setup is not new, e.g.[1], the details of how the I->R model and the IR->O model are implemented led to improved results in the 5 datasets of the ERASER benchmark.

During the rebuttal, the authors added new results to quantify the consistency between R and O using their method vs. previous work, and showed greater consistency. This result is an afterthought, so the authors should make sure to add it to the paper.


[1]Wiegreffe et al., Measuring Association Between Labels and Free-Text Rationales.